# Development of an LC-MS/MS Method for ARV-110, a PROTAC Molecule, and Applications to Pharmacokinetic Studies

**DOI:** 10.3390/molecules27061977

**Published:** 2022-03-18

**Authors:** Thi-Thao-Linh Nguyen, Jin Woo Kim, Hae-In Choi, Han-Joo Maeng, Tae-Sung Koo

**Affiliations:** 1College of Pharmacy, Gachon University, 191 Hambakmoe-ro, Yeonsu-gu, Incheon 21936, Korea; linh.nguyen@gachon.ac.kr; 2Graduate School of New Drug Discovery and Development, Chungnam National University, Daejeon 34134, Korea; dpslzk333@naver.com (J.W.K.); chi705@naver.com (H.-I.C.)

**Keywords:** ARV-110, proteolysis-targeting chimera, LC-MS/MS, validation, stability, pharmacokinetics

## Abstract

ARV-110, a novel proteolysis-targeting chimera (PROTAC), has been reported to show satisfactory safety and tolerability for prostate cancer therapy in phase I clinical trials. However, there is a lack of bioanalytical assays for ARV-110 determination in biological samples. In this study, we developed and validated an LC-MS/MS method for the quantitation of ARV-110 in rat and mouse plasma and applied it to pharmacokinetic studies. ARV-110 and pomalidomide (internal standard) were extracted from the plasma samples using the protein precipitation method. Sample separation was performed using a C18 column and a mobile phase of 0.1% formic acid in distilled water–0.1% formic acid in acetonitrile (30:70, *v*/*v*). Multiple reaction monitoring was used to quantify ARV-110 and pomalidomide with ion transitions at m/z 813.4 → 452.2 and 273.8 → 201.0, respectively. The developed method showed good linearity in the concentration range of 2–3000 ng/mL with acceptable accuracy, precision, matrix effect, process efficiency, and recovery. ARV-110 was stable in rat and mouse plasma under long-term storage, three freeze-thaw cycles, and in an autosampler, but unstable at room temperature and 37 °C. Furthermore, the elimination of ARV-110 via phase 1 metabolism in rat, mouse, and human hepatic microsomes was shown to be unlikely. Application of the developed method to pharmacokinetic studies revealed that the oral bioavailability of ARV-110 in rats and mice was moderate (23.83% and 37.89%, respectively). These pharmacokinetic findings are beneficial for future preclinical and clinical studies of ARV-110 and/or other PROTACs.

## 1. Introduction

As a promising and appealing technology for targeted protein degradation, proteolysis-targeting chimeras (PROTACs) offer a chemical knockdown strategy for a protein of interest (POI). The PROTAC approach was first proposed in 2001 by Sakamoto et al. [1] and remarkable developments have been made over the last 20 years [2,3]. Because PROTAC technology enables the elimination of the entire POI, it can potentially overcome the resistance to current treatments. In addition, as PROTACs act catalytically, the required doses of these compounds are much lower than those of conventional small molecules. Owing to their low susceptibility to the increased expression of POIs, PROTACs show impressive inhibition activity against some drug-resistant targets such as the androgen receptor (AR), the estrogen receptor, and Bruton’s tyrosine kinase [4]. Other advantages of PROTACs include the potential to target undruggable targets, such as the signal transducer and activator of transcription 3 (STAT-3), and the ability to influence non-enzymatic functions by degrading entire proteins [5].

PROTACs are heterobifunctional compounds consisting of a ligand of POI connected to a ligand of an E3 ubiquitin ligase through a linker. This molecular structure promotes the formation of a ternary complex of the POI, PROTAC, and E3, followed by an ubiquitination of POI. Subsequently, the ubiquitinated POI is recognized and degraded by endogenous 26S proteasome in eukaryotic cells [6]. There are more than 600 E3 ubiquitin ligases. However, only several E3 ligases with small-molecule ligands have been used to develop PROTACs, including the Von Hippel–Lindau disease tumor suppressor protein (VHL), cereblon (CRBN), the mouse double minute 2 homologue (MDM2), the inhibitor of apoptosis proteins, and the Skp1-Cullin-F box complex containing Hrt1 [7].

Many PROTACs have been validated in preclinical stages [2]. ARV-110 (bavdegalutamide), was the first PROTAC to enter clinical trials in 2019, which targets AR for the treatment of metastatic castration-resistant prostate cancer (mCRPC) [7]. Initially, the structure of ARV-110 was speculated to contain an enzalutamide ligand and a VHL E3 ligase ligand [6]. However, the structure of ARV-110 was subsequently revealed to consist of a specific AR ligand and a CRBN E3 ligand [8]. ARV-110 effectively degrades both wild-type AR and anti-androgen-therapy-induced AR mutants after oral administration. In various prostate cancer cell lines, ARV-110 degrades 95–98% of AR. Furthermore, compared with enzalutamide, ARV-110 exhibits similar efficacy at lower doses in wild-type AR models. In resistance models, ARV-110 reduces tumor growth by 70–100% [6,9]. Preliminary data from a phase I clinical trial indicated that ARV-110 showed satisfactory safety and tolerability in patients [10]. ARV-110 is currently in a phase II clinical trial (NCT03888612).

As ARV-110 is a potential candidate for mCRPC treatment, various clinical studies should be conducted, including dose-dependent pharmacokinetic and bioequivalence studies. However, as no quantitative bioanalytical assays are currently available for ARV-110 determination in biological samples such as plasma, it is necessary to develop rapid and sensitive methods for ARV-110 quantitation. In the present study, we developed and validated a bioanalytical method for the determination of ARV-110 levels in mouse and rat plasma using LC-MS/MS. A protein precipitation method was employed to prepare plasma samples for analysis due to its advantages of simplicity, cost-effectiveness, minimal samples loss, and feasibility of automation [11]. In addition, we applied the developed method to evaluate the in vitro metabolic stability of ARV-110 and the pharmacokinetic profiles of ARV-110 following oral (PO) and intravenous (IV) administration to mice and rats.

## 2. Results and Discussion

### 2.1. LC-MS/MS Method Development

The chemical structures of ARV-110 and pomalidomide (internal standard, IS) are shown in Figure 1. ARV-110 and IS were protonated in positive ESI mode, and [M + H]^+^ at *m*/*z* 813.4 and 273.8 were selected as the precursor ions of ARV-110 and IS, respectively. The most prominent fragment ions of ARV-110 and IS in the product ion spectra were at *m*/*z* 452.2 and 201.0, respectively (Figure 1). The product ion of ARV-110 was formed by breakage of the C–N linkage (Figure 1a). The product ion of IS was attributed to the breakdown of the piperidine-2,6-dione ring (Figure 1b), similar to a previous report [12]. Therefore, the multiple reaction monitoring (MRM) transitions of 813.4 → 452.2 and 273.8 → 201.0 were selected for ARV-110 and IS, respectively.

For ARV-110, the optimized declustering potential, collision energy, and collision cell exit potential were 151, 55, and 8 V, respectively, whereas those for IS were 71, 29, and 14 V, respectively. Various LC conditions were tested to obtain ARV-110 and IS peaks with adequate retention times and separation. Under the optimized conditions, ARV-110 and IS were separated on an Agilent Zorbax C18 column (5 µm, 4.6 × 150 mm) eluted with 0.1% formic acid in water–0.1% formic acid in acetonitrile (30:70, *v*/*v*)–under isocratic conditions (5 min) at a flow rate of 0.35 mL/min. These conditions resulted in good chromatographic resolution with sharp ARV-110 and IS peaks at 3.7 and 4.1 min, respectively.

### 2.2. Method Validation

With a signal-to-noise ratio of more than 10:1, a lower limit of quantification (LLOQ) of 2 ng/mL was set for both rat and mouse plasma. As shown in Figure 2 and Figure 3, ARV-110 and IS were successfully separated from the endogenous components of blank rat/mouse plasma using the optimized LC-MS/MS conditions. Symmetrical peaks were obtained for ARV-110 and IS at 3.7 and 4.1 min, respectively. Moreover, constant retention times were observed with repeated analysis, which were identical to those observed in the plasma samples from the pharmacokinetic studies for both rat and mouse plasma. These observations confirm the specificity of the developed LC-MS/MS method for ARV-110 analysis in rat and mouse plasma without interference from endogenous substances.

The calibration curves for ARV-110 in rat and mouse plasma showed good linearity over the range of 2–3000 ng/mL with a weighting factor of 1/x^2^. The linear equations for rat and mouse plasma were y = 0.001310x + 0.000042 (r = 0.9921) and y = 0.00124x + 0.00017 (r = 0.9912), respectively.

Quality control (QC) samples of ARV-110 in rat and mouse plasma, including LLOQ (2 ng/mL); low QC (LQ, 5 ng/mL); middle QC (MQ, 100 ng/mL); and high QC (HQ, 2500 ng/mL), were investigated for the accuracy and precision of the proposed method. The relative error (%RE) and the coefficient of variation (CV%) were used to evaluate the accuracy and precision, respectively. In rat plasma, the intra- and interday accuracy values were ≤5.333% and ≤10.61%, respectively, and the intra- and interday precision values were ≤12.60% and ≤9.972%, respectively (Table 1). These values were within acceptable limits (i.e., 20% for LLOQ and 15% for other concentrations), indicating that the ARV-110 analysis method was reproducible and reliable in rat plasma [13,14]. In mouse plasma, the intraday accuracy and precision values were ≤2.333% and ≤12.55%, respectively. These results suggest that the developed method is also reliable for ARV-110 analysis in mouse plasma.

The matrix effect, process efficiency, and recovery of ARV-110 and IS in rat and mouse plasma samples are presented in Table 2. The matrix effect values for ARV-110 and IS were less than 100%, leading to process efficiency values of 70–80%. However, the true recovery value of the extraction method, which was not affected by the matrix effect, was approximately 100%. Moreover, the matrix effect, process efficiency, and recovery values of ARV-110 and IS were consistent and reproducible for all the investigated concentrations. The CV% of the matrix effect met the required criteria of <15% at all QC levels for ARV-110 and IS (2.23–9.11% for rat plasma and 3.45–6.50% for mouse plasma). After normalization with IS, the matrix effect, process efficiency, and recovery values in rat plasma were 87.06–104.06%, 93.21–101.90%, and 98.13–107.87%, respectively, whereas those in mouse plasma were 92.29–104.83%, 102.54–114.83%, and 106.51–111.13%, respectively. These results suggest that the simple protein precipitation method is suitable for the efficient extraction of ARV-110 and IS from rat and mouse plasma. Since the instrument was possibly contaminated by the matrix components of plasma, a sample clean-up was performed prior to and after each sample injection using methanol 50% as the washing solvent. In addition, the curtain plate surface of the mass spectrometer was washed with methanol 50% to remove any retained contaminants prior to starting following projects.

ARV-110 was stable in rat and mouse plasma during long-term storage (4 weeks at −20 °C), over three freeze–thaw cycles, in an autosampler at 10 °C for 24 h, and at room temperature (25 °C) for 15 min (Table 3). Notably, ARV-110 was unstable in rat and mouse plasma during short-term storage (1 h at 25 °C). After 1 h, approximately 69% and 80% of ARV-110 remained in QC samples in rat and mouse plasma, respectively. Because of this behavior, we extensively investigated the plasma and microsomal metabolic stabilities of ARV-110, as described in Section 2.3.

### 2.3. In Vitro Stability

In plasma samples, the amount of remaining ARV-110 rapidly decreased after incubation for 4 h at 37 °C, as shown in Figure 4a, with *T*_1/2_ values of 44, 93, and 24 min in rat, mouse, and human plasma, respectively. The instability of ARV-110 in plasma is likely attributable to hydrolysis by plasma enzymes, which is commonly observed for compounds with amide functional groups [15], particularly glutarimide-containing compounds (e.g., thalidomide) [16]. However, this degradation could be prevented by storing the plasma samples at low temperatures, as ARV-110 was stable in ice-cold rat and mouse plasma for 4 h (Figure 4b) and at −20 °C for 4 weeks (long-term stability test of QC samples, Table 3).

ARV-110 was stable in all tested hepatic microsomal fragments, with 96.60%, 80.95%, and 92.07% of the drug remaining after incubation for 2 h and elimination half-life (*T*_1/2_) values of 2215, 415, and 986 min for rats, mice, and humans, respectively (Figure 4b). These results suggest that the major elimination route of ARV-110 is not phase 1 metabolism by enzymes such as hepatic CYP450.

### 2.4. In Vivo Pharmacokinetic Studies

The developed LC-MS/MS method was applied to pharmacokinetic studies in rats and mice. The mean plasma concentration–time profiles of ARV-110 after IV (2 mg/kg) and PO (5 mg/kg) administration to rats and mice are shown in Figure 5, and the pharmacokinetic parameters are presented in Table 4. In rats, after IV administration of ARV-110, the calculated total clearance (*CL*) value was 413.6 ± 31.7 mL/h/kg, which is quite low compared with the hepatic blood flow rate of rats (55 mL/min/kg, equivalent to 3300 mL/h/kg) [17]. A moderate to high volume of distribution at the steady state (*V_ss_*) value (5775 ± 320 mL/kg) indicated that ARV-110 was well distributed in the tissues. These composite results provide an explanation for ARV-110 existing in rat plasma for 48 h in vivo after IV and PO administration, despite its instability in rat plasma at 37 °C in vitro. Following PO administration of ARV-110, a plasma peak concentration (*C_max_*) value of 110.5 ± 9.2 ng/mL was obtained at 5.5 ± 1.9 h. In addition, the time to reach *C**_max_* (*T*_1/2_) was relatively long (13.62 ± 1.43 and 17.67 ± 3.21 h for IV and PO administration, respectively). The oral bioavailability of ARV-110 in rats was moderate (23.8%).

Upon IV administration in mice, the *CL* value of ARV-110 was low (180.9 ± 30.79 mL/h/kg) as compared to the hepatic blood flow rate of mice (90 mL/min/kg [17], equivalent to 5400 mL/h/kg). In addition, ARV-110 showed a relatively large *V_ss_* value (2366 ± 402.2 mL/kg), indicating that the drug was confined mainly to the tissues. Another PROTAC (AZ-X) has also been reported to show a low *CL* value (16 mL/min/kg) and a moderate to high *V_ss_* value (5200 mL/kg) [18]. Following PO administration, the *C_max_* value was 612.0 ± 88.38 ng/mL at 4.8 ± 1.8 h. Relatively long *T*_1/2_ values (11.41 ± 0.505 and 14.57 ± 2.479 h for IV and PO administration, respectively) were also observed in mice. The oral bioavailability of ARV-110 in mice after dose normalization was 37.9%. Interestingly, in the IV pharmacokinetic profile, a secondary peak emerged at 4–8 h, which was probably due to enteric reabsorption. Although not commonly reported, this phenomenon has been observed for several drugs after IV administration [19].

Despite the high molecular weight of ARV-110, it showed moderate bioavailability in rats and mice. This bioavailability is thought to be due to the introduction of a thalidomide derivative CRBN E3 ligand, which is expected to improve the poor permeation properties of PROTACs [18].

The pharmacokinetic data confirmed that the developed LC-MS/MS assay was appropriate for pharmacokinetic studies and relevant investigations of ARV-110. To the best of our knowledge, this is the first report of in vivo pharmacokinetic parameters after IV and PO administration of ARV-110 in rats and mice. The results are expected to be helpful for further preclinical and clinical studies on ARV-110.

## 3. Materials and Methods

### 3.1. Materials

ARV-110 (C_41_H_43_ClFN_9_O_6_; MW = 812.29; logP = 3.56; pKa1 = 1.51; pKa2 = 8.02 as predicted by MarvinSketch version 22.7, Chemaxon) was chemically synthesized (Appendix A) and pomalidomide (IS; Cat. No. P2074, logP = 0.01, pKa = 1.89 [20]) was provided by Tokyo Chemical Industry (Tokyo, Japan). HPLC-grade water (Cat. No. 7732-18-5) and acetonitrile (Cat. No. 5-05-8) were obtained from J.T. Baker (Phillipsburg, NJ, USA). Formic acid was purchased from Sigma-Aldrich (St. Louis, MO, USA). All other chemicals were of analytical grade.

### 3.2. Instrumentation and Analytical Conditions

ARV-110 was analyzed using an Agilent 1100 HPLC system (Agilent Technologies, Santa Clara, CA, USA) coupled to an API 4000 triple quadrupole mass spectrometer (AB Sciex, Framingham, MA, USA) via an ESI interface in the positive ionization mode. A Zorbax^®^ C18 column (5 µm, 4.6 × 150 mm, Agilent, Santa Clara, CA, USA) maintained at 40 °C was used to separate ARV-110 and IS from the sample matrix. Separation was performed via isocratic elution using a mobile phase of 0.1% formic acid in distilled water–0.1% formic acid in acetonitrile (30:70, *v*/*v*) at a flow rate of 0.35 mL/min with an injection volume of 10 µL. The autosampler temperature was maintained at 4 °C. The MS parameters were as follows: ion voltage, 5500 V; temperature, 600 °C; curtain gas pressure, 20 psi; nebulizer gas pressure, 50 psi; turbo gas pressure, 50 psi; entrance potential, 10 V; declustering potentials, 151 and 71 V (ARV-110 and IS, respectively); collision energies, 55 and 29 V (ARV-110 and IS, respectively); and collision cell exit potentials, 8 and 14 V (ARV-110 and IS, respectively). The MRM mode was used to quantify the ion transitions at *m*/*z* 813.4 → 452.2 and 273.8 → 201.0 for ARV-110 and IS, respectively. The peak areas were integrated automatically using Analyst software version 1.6.2 (Applied Biosystems/MDS SCIEX, Framingham, MA, USA).

### 3.3. Sample Preparation

To prepare the standard samples for the calibration curve, 10-fold concentrated working standard solutions were prepared via serial dilution of an ARV-110 stock solution (10 mg/mL) with acetonitrile. Working standard solutions for the QC samples were prepared independently.

For the rat and mouse plasma samples, 18 µL of blank plasma was spiked with 2 μL of each working standard solution to obtain final ARV-110 concentrations of 2, 3, 10, 100, 1000, 2000, and 3000 ng/mL. To achieve protein precipitation, 20 μL of IS in acetonitrile (10 μg/mL) and 110 μL of acetonitrile were added. After vigorously mixing for 10 min, the mixture was centrifuged at 13,500 rpm for 10 min. Then, 10 μL of the collected supernatant was injected into the LC-MS/MS system. QC samples with ARV-110 concentrations of 2 ng/mL (LLOQ), 5 ng/mL (low QC, LQ), 100 ng/mL (middle QC, MQ), and 2500 ng/mL (high QC, HQ) were prepared similarly. All stock and working standard solutions were stored at −20 °C during the experiments.

### 3.4. Assay Validation

The LC-MS/MS method for ARV-110 analysis in rat plasma was validated in terms of specificity, linearity, accuracy, precision, matrix effect, recovery, process efficiency, and stability according to the guidelines of the European Medicines Agency and the United States Food and Drug Administration [13,14]. Following full validation in rat plasma, the method was partially validated for ARV-110 analysis in mouse plasma (the same validation parameters as determined in rat plasma, except for the interday accuracy and precision).

The specificity was evaluated using six lots of untreated blank rat and mouse plasma. The chromatograms of blank plasma, blank plasma spiked with IS only, blank plasma spiked with ARV-110 and IS, and plasma from pharmacokinetic samples were compared. The retention times of the analyte and IS were compared to confirm the absence of interference peaks from endogenous substances in the plasma.

The calibration curves of ARV-110 in plasma over the concentration range of 2–3000 ng/mL were obtained by plotting the analyte-to-IS peak area ratios against the nominal concentrations of the standards. The calibration curves were fitted using least-squares linear regression with a weighted factor of 1/x^2^. Linearity was assessed using the correlation coefficients (r), with values ≥0.990 considered acceptable [21].

The accuracy and precision of the analytical method were evaluated using the LLOQ (2 ng/mL) and QC samples (5, 100, and 2500 ng/mL). The intraday accuracy and precision were determined by repeating the experiments six times per day, whereas the interday data were obtained by repeating the experiments for three days. The %RE, calculated as the difference between the calculated and nominal concentrations divided by the nominal concentration of ARV-110 and expressed as a percentage, was used to assess the accuracy. The precision was assessed using CV%, which was calculated as the SD of the peak area ratio divided by the mean peak area ratio of ARV-110 and IS, expressed as a percentage.

The matrix effect, extraction recovery, and process efficiency were evaluated for each QC group. The matrix effect indicates whether the endogenous components of the sample matrix affect the ionization of the analyte and IS. The extraction recovery and process efficiency were used to assess the ability of the sample extraction method to consistently recover the analyte and IS. The former represents the true recovery value, which is not affected by the matrix effect, whereas the latter shows the overall process efficacy [22]. The matrix effect was calculated by dividing the mean peak areas of ARV-110 spiked in the extracted blank plasma (set 2) by the peak area of the analyte in acetonitrile (set 1). The recovery was evaluated by comparing the mean peak areas of ARV-110 in the sample extracts (set 3) with those of the set 2 samples. The process efficiency was calculated by comparing the data in sets 3 and 1 [23,24].

The sample stability in rat and mouse plasma was evaluated at low and high QC concentrations. The short-term stability was evaluated at room temperature for 1 h, whereas the long-term stability was determined by assaying samples stored at −20 °C for 4 weeks. The processed sample stability was evaluated using samples prepared and kept in an autosampler at 4 °C for 24 h before analysis. The freeze–thaw stability was tested by subjecting samples to three freeze–thaw cycles.

### 3.5. In Vitro Metabolic Stability

The stabilities of ARV-110 in the hepatic microsomal fraction and plasma were investigated for three species (human, rat, and mouse). For the microsomal stability study, a reaction mixture containing hepatic microsomes (protein concentration, 0.5 mg/mL) and 1 mM NADPH in phosphate buffer (pH 7.4) was pre-incubated in a thermomixer (5 min, 37 °C, 200 rpm). To initiate the metabolic reaction, 5 μL of ARV-110 in methanol (100 μM) was added to the pre-incubated mixture to obtain a final ARV-110 concentration of 1 μM. The metabolic reactions of buspirone and verapamil were conducted simultaneously as positive controls. For plasma stability measurements, 5 μL of ARV-110 in methanol (100 μM) was added to pre-warmed blank plasma to obtain a final ARV-110 concentration of 1 μM. Procaine was used to control plasma stability. The temperature was maintained and agitation was continued throughout the reaction, and sampling was conducted at predetermined time points (0 (beginning of the reaction), 15; 30; 60; 120; 360; and 480 min). For the microsomal and plasma stability measurements, the reaction was conducted for 120 and 240 min, respectively. At each sampling point, 50 μL of the reaction mixture was transferred to a 1.5 mL microtube containing 100 μL of ice-cold IS in methanol, followed by vortex mixing to terminate the reaction. The mixture was then centrifuged at 16,000 rpm for 15 min at 4 °C, and the supernatant was injected into the LC-MS/MS system to determine the amount of analyte remaining. The elimination rate constant (*k_e_*) was determined as the slope of the linear regression between the log of the percentage of analyte remaining and the incubation time. The *T*_1/2_ value was estimated using the following equation [25,26]: *T*_1/2_ = ln2/*k_e_*. The experiments were conducted in triplicate, and data are presented as mean ± SD.

### 3.6. Application in Pharmacokinetics Studies

The validated LC-MS/MS method was applied to determine ARV-110 in pharmacokinetic studies in rats and mice. Animal studies were approved in advance by the Institutional Animal Care and Use Committee of Chungnam National University (202103A-CNU-053; Daejeon, Republic of Korea). Male Sprague Dawley (SD) rats (7 weeks old, 210–251 g) and male ICR mice (6 weeks old, 25–32 g) were purchased from Orient Bio Inc. (Seongnam, Republic of Korea). Their habitat was maintained under a 12 h light–dark cycle at a room temperature of 20–25 °C and a relative humidity of 40–60%. The animals had free access to food and water. Food fasting was started 14 or 4 h prior to drug administration to rats and mice, respectively. After drug administration, the animals were not given food or water for another 4 h.

The drug solution for the pharmacokinetic study in rats was prepared by dissolving ARV-110 in a dimethyl sulfoxide (DMSO)–PEG400–Cremophor EL–saline (5:40:5:50, *v*/*v*/*v*/*v*). The drug solution was administered to the rats via a single IV bolus into the tail vein at a dose of 2 mg/kg or PO using a gavage needle at a dose of 5 mg/kg. Blood samples (100 μL) were collected from the jugular vein using heparinized syringes at 0.083 (IV only), 0.25, 0.5, 1, 2, 4, 6, 8, 24, and 48 h after ARV-110 administration. Plasma was collected from the blood by centrifugation, and a 20 μL aliquot of plasma was stored at −20 °C until analysis.

The drug solution for the pharmacokinetic study in mice was prepared by dissolving ARV-110 in DMSO–PEG400–Cremophor EL–saline (5:30:2.5:62.5, *v*/*v*/*v*/*v*). The drug solution was administered via an IV bolus into the tail vein at a dose of 2 mg/kg or PO at a dose of 5 mg/kg. Blood samples (50 μL) were alternatively collected from the right retro-orbital plexus at 0.05 (IV only), 0.12, 0.5, 1, 2, 4, 8, 24, and 48 h after ARV-110 administration. Plasma was obtained by centrifuging the blood at 17,600× *g* for 5 min, and a 20 μL aliquot of plasma was stored at −20 °C until analysis.

For plasma sample preparation, 20 μL of rat or mouse plasma was added to 20 μL of IS solution (10 μg/mL) and 110 μL of acetonitrile. After vigorous mixing for 10 min, the mixture was centrifuged at 13,500 rpm for 10 min. An aliquot of the supernatant (100 μL) was collected for analysis. *C**_max_* and *T**_max_* were determined directly from the plasma concentration–time profile of each animal. To determine other pharmacokinetic parameters, the experimental data were analyzed using Phoenix^®^ 8.2 software (Certara L.P., Princeton, NJ, USA), as described previously [23,27]. All data are presented as mean ± SD.

## 4. Conclusions

In this study, an LC-MS/MS method for the quantitation of ARV-110 in rat and mouse plasma was developed and validated according to the guidelines of the European Medicines Agency and the United States Food and Drug Administration. The developed bioanalytical method showed good reproducibility and reliability. This method was successfully applied to the pharmacokinetic study of ARV-110 in both rats and mice, revealing a moderate oral bioavailability for this drug. In addition, ARV-110 showed good metabolic stability in rat, mouse, and human hepatic microsomal fractions. In contrast, the drug was unstable in rat, mouse, and human plasma at room temperature and 37 °C, although it was stable under cold conditions. The developed analysis method and the findings of this study will be helpful for further investigations and clinical studies of ARV-110.

## Figures and Tables

**Figure 1 molecules-27-01977-f001:**
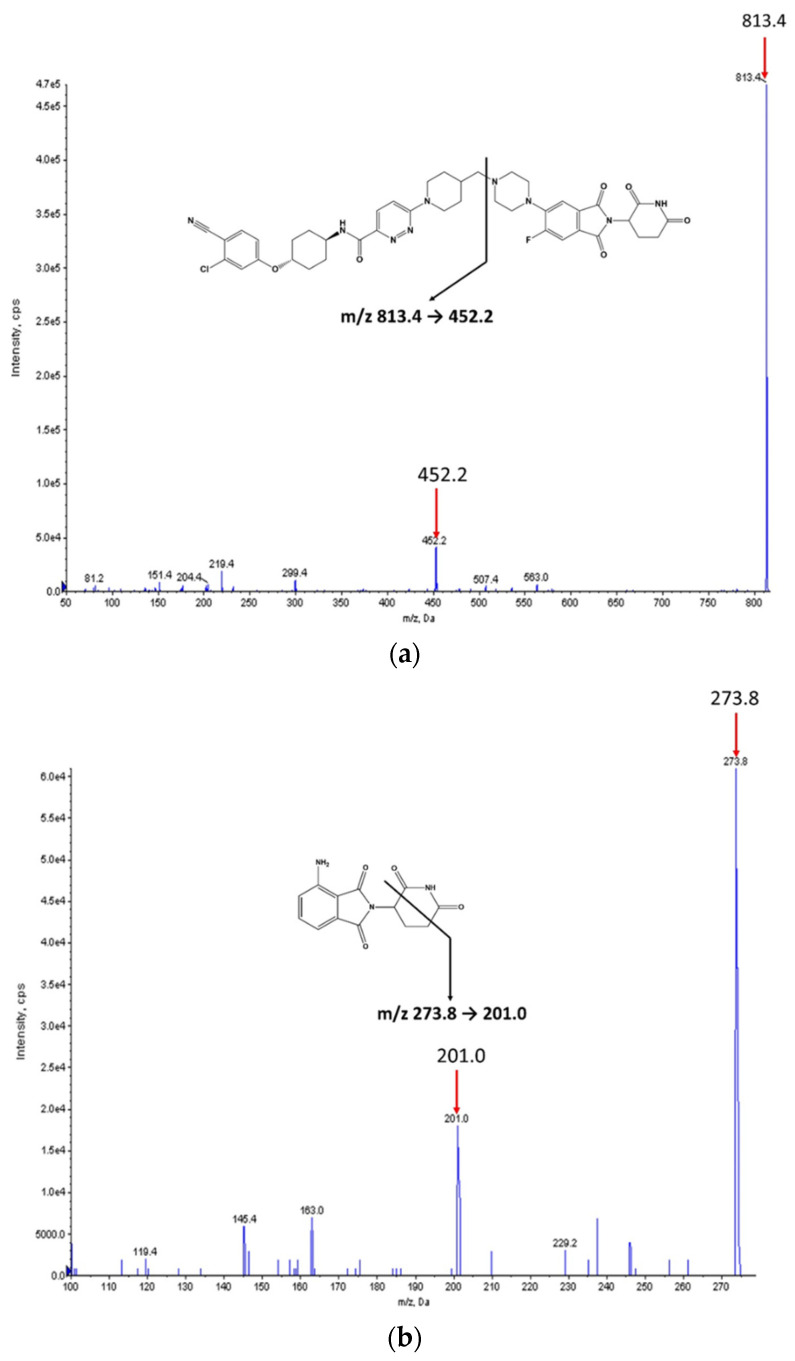
Representative product ion spectra of ARV-110 (**a**); and pomalidomide (**b**) in positive ionization mode.

**Figure 2 molecules-27-01977-f002:**
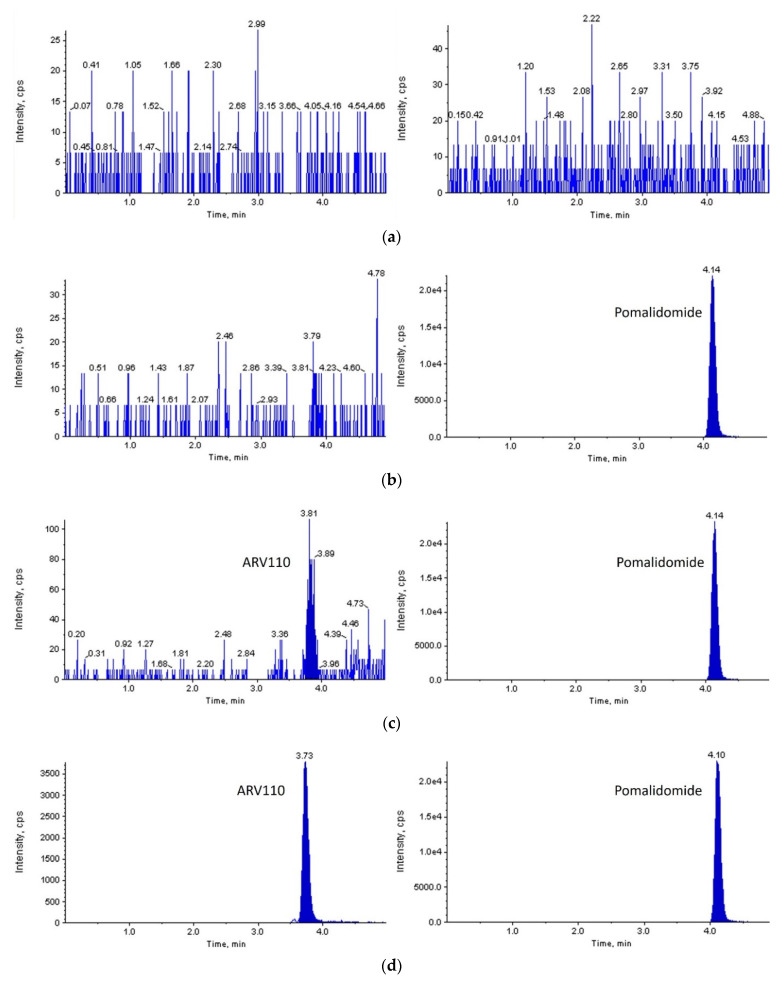
MRM LC-MS/MS chromatograms of ARV-110 (left) and pomalidomide (right) after deproteinization of blank rat plasma (**a**); zero calibrator (**b**); LLOQ (**c**); and rat plasma sample 60 min after PO administration of ARV-110 at a dose of 2 mg/kg (**d**).

**Figure 3 molecules-27-01977-f003:**
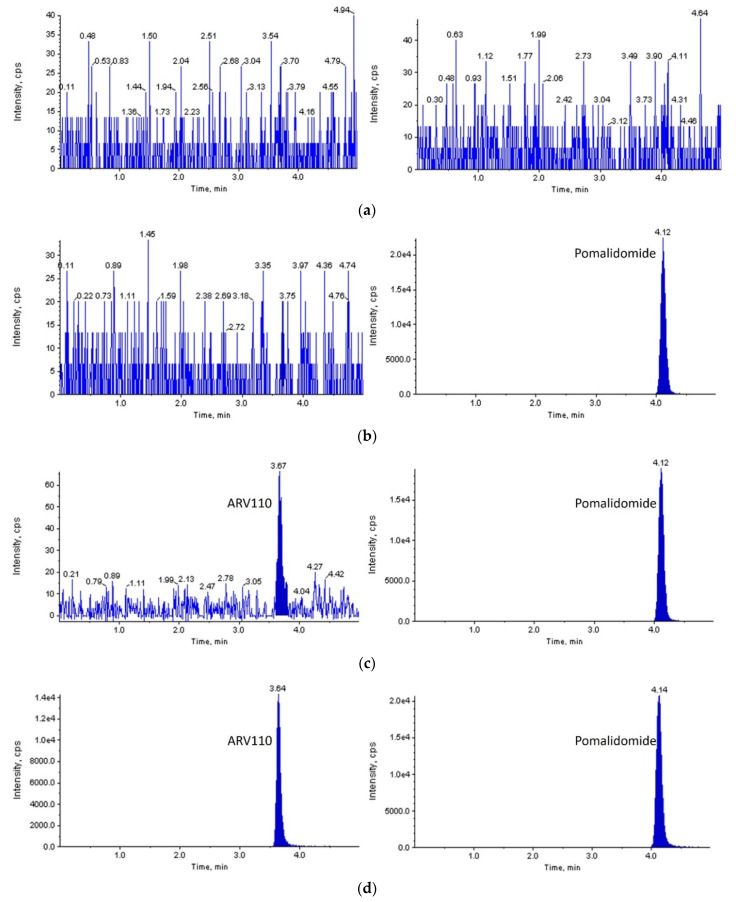
MRM LC-MS/MS chromatograms of ARV-110 (left) and pomalidomide (right) after deproteinization of blank mouse plasma (**a**); zero calibrator (**b**); LLOQ (**c**); and mouse plasma sample 60 min after PO administration of ARV-110 at a dose of 2 mg/kg (**d**).

**Figure 4 molecules-27-01977-f004:**
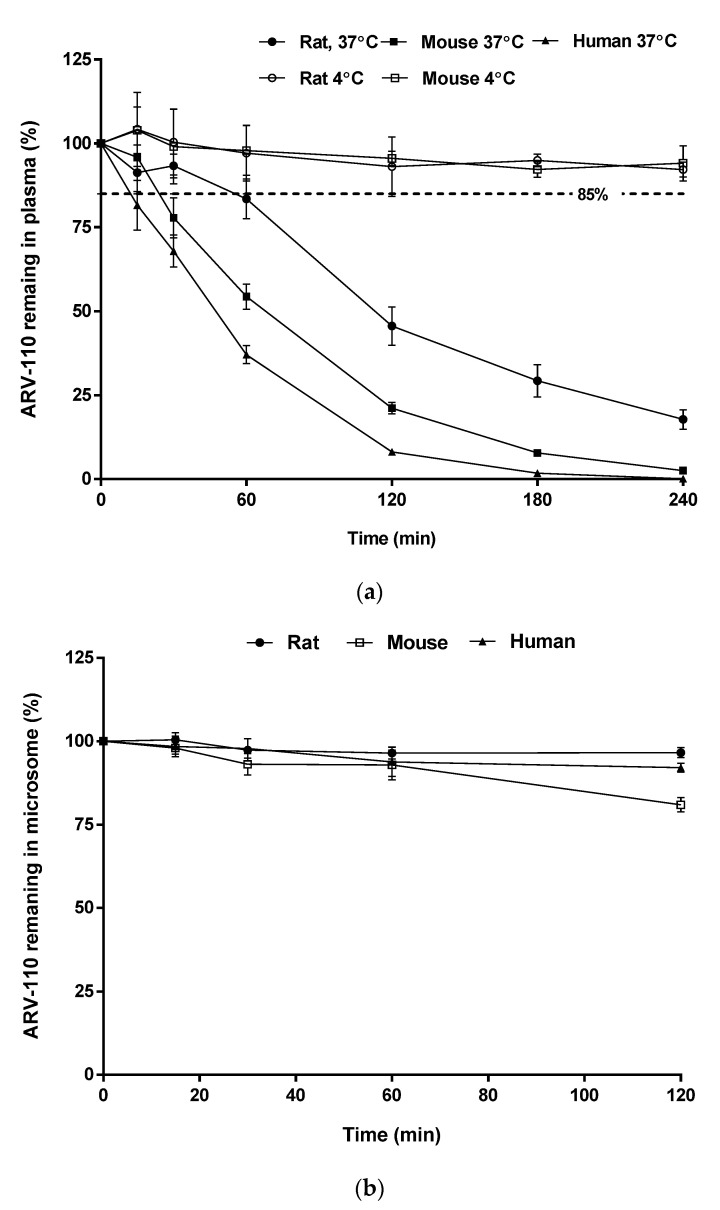
Residual ARV-110 levels (%) in rat plasma at 4 °C (○), rat plasma at 37 °C (●), mouse plasma at 4 °C (□), mouse plasma at 37 °C (■) and human plasma at 37 °C (▲) (**a**); and in rat (●), mouse (□), and human (▲) hepatic microsomes (**b**). Data are presented as mean ± SD (*n* = 3).

**Figure 5 molecules-27-01977-f005:**
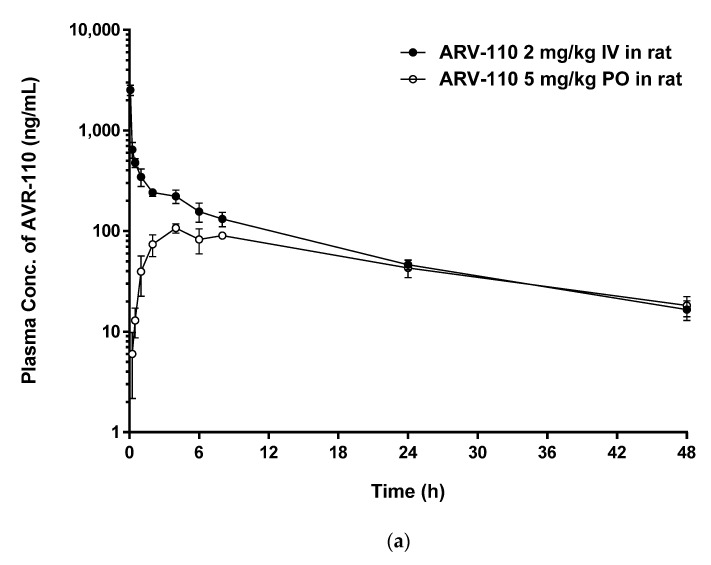
Plasma concentration–time curves after IV administration at 2 mg/kg (●) and PO administration at 5 mg/kg (○) to rats (**a**); and mice (**b**). Data are presented as mean ± SD (*n* = 4).

**Table 1 molecules-27-01977-t001:** Intra- and interday precision and accuracy of the assay.

Nominal Concentration (ng/mL)	Measured Concentration (ng/mL)	Accuracy (%RE)	Precision (%CV)
Rat plasma			
Intraday (*n* = 6)			
2	2.107 ± 0.265	5.333	12.60
5	4.900 ± 0.306	2.000	6.240
100	103.1 ± 9.426	3.083	9.144
2500	2608 ± 196.1	4.333	7.518
Interday (*n* = 18)			
2	2.000 ± 0.199	0.139	9.972
5	5.290 ± 0.385	5.700	7.278
100	110.6 ± 8.540	10.61	7.721
2500	2649 ± 202.9	5.956	7.662
Mouse plasma			
Intraday (*n* = 6)			
2	2.047 ± 0.257	2.333	12.55
5	4.867 ± 0.370	2.667	7.603
100	99.20 ± 8.536	0.850	8.609
2500	2513 ± 249.6	0.533	9.932

**Table 2 molecules-27-01977-t002:** Matrix effect, process efficiency, and extraction recovery of the assay in rat and mouse plasma (*n* = 5).

Nominal Concentration (ng/mL)	Matrix Effect (%)	CV * (%)	Process Efficiency (%)	Recovery (%)
Rat plasma				
ARV-110				
5	67.57 ± 1.51	2.23	70.98 ± 4.18	105.6 ± 7.1
100	80.76 ± 7.35	9.11	77.60 ± 2.91	96.09 ± 3.61
2500	71.36 ± 4.19	5.87	73.60 ± 3.14	103.1 ± 4.4
IS				
Pomalidomide (10 µg/mL)	77.61 ± 3.19	4.11	76.15 ± 2.64	97.92 ± 3.67
Mouse plasma				
ARV-110				
5	69.55 ± 4.52	6.50	70.87 ± 2.33	101.9 ± 3.4
100	79.01 ± 2.73	3.45	79.43 ± 4.56	100.5 ± 5.8
2500	73.33 ± 3.29	4.48	71.62 ± 9.64	97.66 ± 13.14
IS				
Pomalidomide (10 µg/mL)	75.37 ± 3.19	4.24	69.11 ± 3.72	91.69 ± 4.93

*: coefficient of variation of the matrix effect.

**Table 3 molecules-27-01977-t003:** Stability of ARV-110 in rat and mouse plasma (mean ± SD, *n* = 4).

Storage Conditions	Nominal Concentration (ng/mL)	Stability in Rat Plasma (%)	Stability in Mouse Plasma (%)
Processed sample (autosampler, 10 °C, 24 h)	5	99.60 ± 2.91	98.77 ± 3.58
	2500	102.6 ± 4.1	98.62 ± 3.54
Long term (4 weeks at −20 °C)	5	100.6 ± 5.6	97.97 ± 10.23
	2500	100.5 ± 5.0	97.59 ± 3.78
Freeze–thaw (3 cycles)	5	103.2 ± 3.9	104.0 ± 6.1
	2500	101.8 ± 3.4	103.2 ± 4.1
15 min at room temperature (25 °C)	5	100.6 ± 7.8	99.43 ± 2.93
	2500	96.99 ± 2.27	95.74 ± 5.74
1 h at room temperature (25 °C)	5	68.73 ± 7.74	83.70 ± 14.24
	2500	68.82 ± 1.08	78.91 ± 4.93
4 h on ice cold	5	92.20 ± 1.38	93.01 ± 5.30
	2500	92.21 ± 2.24	94.09 ± 5.23

**Table 4 molecules-27-01977-t004:** Pharmacokinetic parameters of ARV-110 after IV and PO administration to rats and mice (*n* = 4).

Parameter	Rat		Mouse	
	Intravenous	Oral	Intravenous	Oral
*T_max_* (h)	0.083 ± 0.000	5.500 ± 1.915	0.050 ± 0.000	4.800 ± 1.789
*C_max_* (ng/mL)	2525 ± 292	110.5 ± 9.2	1263 ± 99	612.0 ± 88.38
*T*_1/2_ (h)	13.62 ± 1.43	17.67 ± 3.21	11.41 ± 0.51	14.57 ± 2.479
*AUC_last_* (ng∙h/mL)	4527 ± 288	2417 ± 222	10,756 ± 1848	9873 ± 1005
*AUC_inf_* (ng∙h/mL)	4857 ± 381	2894 ± 318	11,304 ± 1964	10,707 ± 1176
*CL* (mL/h/kg)	413.6 ± 31.7	-	180.9 ± 30.8	-
*MRT* (h)	10.16 ± 0.48	16.41 ± 1.16	13.12 ± 0.90	12.89 ± 0.84
*V_ss_* (mL/kg)	5775 ± 320	-	2366 ± 402	-
Bioavailability (%)	-	23.83 ± 2.62	-	37.89 ± 4.16

## Data Availability

The data presented in this study are available in the article.

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
