# Peer review of "Development of an LC-MS/MS Method for ARV-110, a PROTAC Molecule, and Applications to Pharmacokinetic Studies"

_molecules, 2022, doi:10.3390/molecules27061977_

Round 1

Reviewer 1 Report

The manuscript entitled “Development of an LC-MS/MS method for ARV-110, a PROTAC molecule, and applications to pharmacokinetic studies” provides a very interesting study on the development and validation of a robust LC-MS/MS method for the quantification of ARV-110, which is a proteolysis targeting chimera (PROTAC) currently under phase II clinical investigation. Beside an important and detailed description of the developed method parameters, the study presents also relevant data on in vitro and in vivo pharmacokinetic (PK) properties of ARV-110, never reported so far.   

Overall, the work performed is clearly justified, well-structured, and very well written, with the exception of only a small section that needs to be revised in its presentation. Thus, I support the publication of the manuscript after minor revisions described below.

  • Page 1, line 37: At least one or two reviews collecting the most promising results achieved with PROTAC technology over the last 20 years should be cited as well.
  • Page 2, line 45: “..activator of transcription 3, and..” What is the meaning of number 3?
  • Page 2, line 49: In “..of a ternary complex by the POI and E3..” please replace ‘by’ with ‘between’.
  • Page 2, line 56: Please cite appropriate references to support the sentence.
  • Page 2, line 67: Please give clinical trial number as reference.
  • Page 2, line 88: Please correct typo ARV-100 with ARV-110.
  • Page 2, line 93: ‘in acetonitrile’ is lacking before (30:70, v/v).
  • Table 1 is never cited in the text along all the manuscript. Please insert it appropriately.
  • Page 7, lines 147-152 and 2.3. section. The results described in these few lines are very interesting but their presentation given by authors is very confusing. I suggest different modifications: 1) Authors should include also this small paragraph (lines 147-152) under the 2.3. “in vitro stability” section. 2) Authors should better describe the experiments performed by harmonizing data described in the text with those reported in Table 3. Indeed, there are data mentioned in the text not reported in Table 3 (rat and mouse plasma stability at 1h at 25°C) and vice versa (15 min rt and 4h ice). 3) For each storage condition reported in Table 3, please give the same info about temperature and exposure time. 4) To report clearer the data, I suggest authors to describe first all the results they obtained on the in vitro plasma stability and then describe in vitro metabolic stability. This means that lines 160-162 should be shifted before lines 156-160. Accordingly, Figure 4B should be presented before Figure 4A.
  • Page 7, lines 163-4. Further hypothesis can be postulated. Authors should cite different papers supporting the known chemical instability to hydrolysis of glutarimide-containing compounds.
  • Page 10, line 207. Is reference 13 appropriate?
  • Material and Method: 1) Authors stated that “ARV-110 was chemically synthesized” but no appropriate reference is given for synthesis details. 2) Can the final percentage of DMSO maintained in the tested samples of the different experiments be better clarified?

Author Response

In response to Reviewer #1

  1. Page 1, line 37: At least one or two reviews collecting the most promising results achieved with PROTAC technology over the last 20 years should be cited as well.

    : Thanks for the reviewer’s insightful comments. According to the reviewer’s suggestions, we have cited 2 references as follows (Please see the new Ref. #2 and #3 in the revised manuscript).

2. Békés, M.; Langley, D.R.; Crews, C.M. PROTAC targeted protein degraders: the past is prologue. Nature Reviews Drug Discovery 2022, 10.1038/s41573-021-00371-6, doi:10.1038/s41573-021-00371-6.

3. Zhong, Y.; Chi, F.; Wu, H.; Liu, Y.; Xie, Z.; Huang, W.; Shi, W.; Qian, H. Emerging targeted protein degradation tools for innovative drug discovery: From classical PROTACs to the novel and beyond. European Journal of Medicinal Chemistry 2022, 231, 114142, doi:https://doi.org/10.1016/j.ejmech.2022.114142.

  1. Page 2, line 45: “..activator of transcription 3, and..” What is the meaning of number 3?

: Signal transducer and activator of transcription 3 is a transcription factor of the STAT protein family.  In humans, it is encoded by the STAT3 gene. And for clarity, abbreviation is now inserted in the revised manuscript (please see line 45).

“… such as the signal transducer and activator of transcription 3 (STAT-3), and…”

  1. Page 2, line 49: In “..of a ternary complex by the POI and E3..” please replace ‘by’ with ‘between’.

: Thanks for the reviewer’s suggestion. The sentence was modified as follows (please see line 49).

“...of a ternary complex of the POI, PROTAC, and E3...”.

  1. Page 2, line 56: Please cite appropriate references to support the sentence.

: Thanks for the reviewer’s suggestion. We have cited Reference #2 to support the statement (please see line 56).

2. Békés, M.; Langley, D.R.; Crews, C.M. PROTAC targeted protein degraders: the past is prologue. Nature Reviews Drug Discovery 2022, 10.1038/s41573-021-00371-6, doi:10.1038/s41573-021-00371-6.

  1. Page 2, line 67: Please give clinical trial number as reference.

: We have added the clinical trial number, NCT03888612, as suggested by the reviewer. Please see line 68 in the manuscript.

  1. Page 2, line 88: Please correct typo ARV-100 with ARV-110.

: We have corrected the typo as suggested by the reviewer. Thank reviewer for the careful review.

  1. Page 2, line 93: ‘in acetonitrile’ is lacking before (30:70, v/v).

: We have added the missing words as suggested by the reviewer (please see line 97).

  1. Table 1 is never cited in the text along all the manuscript. Please insert it appropriately.

: According to the reviewer’s suggestions, we have cited Table 1 in line 129.

  1. Page 7, lines 147-152 and 2.3. section. The results described in these few lines are very interesting but their presentation given by authors is very confusing. I suggest different modifications: 1)Authors should include also this small paragraph (lines 147-152) under the 2.3. “in vitro stability” section. 2) Authors should better describe the experiments performed by harmonizing data described in the text with those reported in Table 3. Indeed, there are data mentioned in the text not reported in Table 3 (rat and mouse plasma stability at 1h at 25°C) and vice versa(15 min rt and 4h ice). 3) For each storage condition reported in Table 3, please give the same info about temperature and exposure time. 4) To report clearer the data, I suggest authors to describe first all the results they obtained on the in vitro plasma stability and then describe in vitro metabolic stability. This means that lines 160-162 should be shifted before lines 156-160. Accordingly, Figure 4B should be presented before Figure 4A.

: Thanks for the reviewer’s insightful comments.

(1) This paragraph is a part of the method validation and was placed under section 2.2-Method validation. Section 2.3-In vitro stability was placed right after this paragraph; therefore, we humbly think that it is unnecessary to repeat it again in section 2.3.

(2) According to the reviewer’s suggestions, we have revised the text and Table 3.

(3) We have corrected Table 3 as suggested by the reviewer.

(4) We have revised section 2.3 and Figure 4 as suggested by the reviewer.

  1. Page 7, lines 163-4. Further hypothesis can be postulated. Authors should cite different papers supporting the known chemical instability to hydrolysis of glutarimide-containing compounds.

: Thanks for the reviewer’s insightful comments. We have added Reference #16 and revised the sentence to support the chemical instability to hydrolysis of glutarimide-containing compounds (please see line 172-173).

  1. Page 10, line 207. Is reference 13 appropriate?

: We made a mistake when citing the reference. The correct one is now Reference #18. Please see line 221 in the revised manuscript.

  1. Material and Method: 1)Authors stated that “ARV-110 was chemically synthesized” but no appropriate reference is given for synthesis details. 2) Can the final percentage of DMSO maintained in the tested samples of the different experiments be better clarified?

: (1) According to the reviewer’s suggestion, we have added NMR and LC-Mass charts to 'Supplementary' so that you can check the information of the provided substance. (2) DMSO was used only in the rat and mouse PK study and was 5% of the administered solution (we mentioned the percentage of DMSO in the manuscript, please see line 344 and 352). Since rats and mice were administered at 2 mL/kg and 10 mL/kg, respectively, in the case of a rat with a blood volume of 54 mL/kg, the DMSO ratio in the blood at intravenous administration is calculated to be 0.185%. In addition, because DMSO is rapidly distributed and cleared from blood, the concentration in the tested sample is expected to be much lower, and therefore, the remaining DMSO is not expected to affect the quantitation of ARV-110.

Reviewer 2 Report

This work describes the development and validation of an LC-MS/MS method for the determination of a cancer treatment drug named “ARV-110”. Although the authors used a language that is a bit unfamiliar for the readers of this journal in the introduction, for which it is highly recommended to make substantial changes, the development and validation seem sufficient. However, there are some major concerns that need to be addressed for the revision. In the view of this reviewer, the protein precipitation method, which is used in this work, is the most common method and does not require a huge amount of method development and can be done in any laboratory setting. What can make this work publishable is the pharmacokinetic and in-vitro studies that are performed in this manuscript.

In the sample preparation procedure for obtaining the calibration curve of ARV-110 in plasma, the authors did not incubate the samples with the analytes to be equilibrated with the plasma matrix. They performed protein precipitation right after spiking. What if there is protein binding which I am sure there is given the structure of the molecule and the number of sites that can interact with proteins via hydrophobic and electrostatic interactions. This may invalidate the whole project. Please explain.

Such study is required especially if authors want to study the stability of the drug in plasma. In other words, they might be stable at room or 37C but just bind with the protein and precipitate and therefore did not show up in the final extract for stability tests. To perform such a study authors should obtain free and total concertation during stability experiments.

If the drug is not stable at 37C how could it be effective for the treatment which it is as mentioned by the authors and it showed some success in clinical trials?

The absolute matrix effect study demonstrates that the matrix can affect the signal. Therefore, one of the main goals in protein precipitation which is quantitation using instrumental signal (with no normalization) is not achievable. Plus, contamination of the instrument is possible due to the extraction of matrix components. Therefore, authors should highlight the essence of a sample clean-up for the following projects. In addition, they should emphasize the fact that one of the procedures to normalize such matrix effect in quantitation of the target analyte must be implemented (matrix-matched calibration, standard addition of the use of an appropriate internal standard)

Please explain what the basis is of selecting pomalidomide as an internal standard because its molecular structure and molecular mass are different from than analyte. Any information on chemical properties?

Chemical properties of analytes and the internal standard must be provided (logP, Pka).

Line 125: please add the acceptable limits by citing the appropriate guidelines in “These values were within acceptable limits, in”

Author Response

In response to Reviewer #2

  1. This work describes the development and validation of an LC-MS/MS method for the determination of a cancer treatment drug named “ARV-110”. Although the authors used a language that is a bit unfamiliar for the readers of this journal in the introduction, for which it is highly recommended to make substantial changes, the development and validation seem sufficient. However, there are some major concerns that need to be addressed for the revision. In the view of this reviewer, the protein precipitation method, which is used in this work, is the most common method and does not require a huge amount of method development and can be done in any laboratory setting. What can make this work publishable is the pharmacokinetic and in-vitro studies that are performed in this manuscript.

: Thanks for the reviewer’s insightful comments. Following the reviewer’s suggestions, we have revised the Introduction section. In addition, the manuscript was edited by a professional English language editing company to make the language suitable for publication. We have added a sentence in the introduction about the advantage of protein precipitation method in samples preparation (please see line 75-77). We agree that the protein precipitation method is the common and can be done in any laboratory setting. However, the primary goal of our study is to develop and validate an LC-MS/MS method for ARV-110 quantitation in mouse and rat plasma. The method development includes sample preparation and LC-MS/MS analysis. Until now, no quantitative bioanalytical assays are currently available for ARV-110 determination in biological samples such as plasma, it is necessary to develop simple, rapid and sensitive methods for ARV-110 quantitation. Therefore, our developed method is essential for further investigations and clinical studies of ARV-110. In addition, the validation is an essential task for any analytical method to be used in laboratory and industrial scale [1,2].

  1. In the sample preparation procedure for obtaining the calibration curve of ARV-110 in plasma, the authors did not incubate the samples with the analytes to be equilibrated with the plasma matrix. They performed protein precipitation right after spiking. What if there is protein binding which I am sure there is given the structure of the molecule and the number of sites that can interact with proteins via hydrophobic and electrostatic interactions. This may invalidate the whole project. Please explain. Such study is required especially if authors want to study the stability of the drug in plasma. In other words, they might be stable at room or 37C but just bind with the protein and precipitate and therefore did not show up in the final extract for stability tests. To perform such a study authors should obtain free and total concertation during stability experiments.

: Thanks for the reviewer’s careful comments. We agree with the reviewer that ARV-110 tends to interact with plasma protein like other hydrophobic compounds, which is called protein binding. The protein binding of a drug to a protein can be viewed as a reversible and rapid equilibrium process governed by the law of mass action [3]. It is known that two plasma proteins are most responsible for the binding of drug molecules, albumin and α1-acid glycoprotein. The binding is primarily by hydrophobic interaction [4]. Although we do not incubate the spiked standard stock solution in plasma, during the sample preparation using protein precipitation (deproteinization) method, both bound and unbound drug (i.e., total form) are extracted to the organic solvent by the deproteinization in the end. Similarly, we have published several LC-MS/MS bioanalytical method validation and development papers with similar sample preparation procedure (i.e., protein precipitation by methanol or acetonitrile)  [5-9].

Similar to calibration samples, for the plasma stability and pharmacokinetic samples, the determination was done for total form (both bound and unbound drug). Thus, we think that the drug-plasma protein binding could not affect the extraction recovery of the whole project.

Nevertheless, based on the reviewer’s suggestions, we have conducted the protein-binding study in rat, mouse, and human plasma. The protein binding profile of ARV-110 in plasma was evaluated by equilibrium dialysis technique using rapid equilibrium dialysis (RED) device with the procedure obtained from the manufacturer (Pierce Biotechnology, Thermo Fisher Scientific Inc., Waltham, MA, USA) [10,11]. An aliquot (200 µL) of the spiked plasma samples and an aliquot (400 µL) of isotonic phosphate buffered saline were placed into the sample chamber and the adjacent chamber, respectively. The device was then incubated at 37 °C with shaking rate at 150 rpm for 2 hours. The drug concentration measured in plasma compartment was the total form of both bound and unbound drug whereas the concentration in buffer side was the free form. Namely, the fraction unbound in plasma (fp) was calculated by dividing the drug concentrations in ‘the buffer’ compartment (Cf) by those of the plasma compartment (Cp). The concentration of drug bound to plasma protein (Cb) was calculated as the difference between Cp and Cf. The protein binding results obtained was over 99.95% of bound ARV-110 in rat, mouse, and human plasma, strongly suggested ARV110 has very high protein binding (%).

Collectively, since during the protein precipitation (deproteinization), both bound and unbound drug (i.e., total form) are extracted to the organic solvent, and total form (both bound and unbound drug) are measured in calibration and unknown plasma stability and pharmacokinetic samples, we humbly think that drug-plasma protein binding could not affect the extraction recovery of the whole project.

  1. If the drug is not stable at 37C how could it be effective for the treatment which it is as mentioned by the authors and it showed some success in clinical trials?

: In our animal pharmacokinetic studies, ARV-110 showed large volume of distribution obtained in both rats and mice models (Table 4), indicating that the drug was well distributed in the tissues in vivo. Although ARV110 has poor stability in plasma, considerable amount of the drug is well distributed, likely resulting in a long half-life.  It is guessed that this pharmacokinetic property (i.e., large tissue distribution) could explain for the effect of treatment and success in clinical trials of ARV-110 despite its instability in plasma at 37 °C.

In addition, from a pharmacokinetic point of view, 'clearance by blood' calculated from blood volume (54 mL/kg in rat) and plasma half-life (44 min) is 51.03 mL/h/kg. This value corresponds to 13% of the total CL value (413.6 mL/h/kg), meaning that only 13% of the drug is degraded by blood during and the rest is cleared by other routes.

  1. The absolute matrix effect study demonstrates that the matrix can affect the signal. Therefore, one of the main goals in protein precipitation which is quantitation using instrumental signal (with no normalization) is not achievable. Plus, contamination of the instrument is possible due to the extraction of matrix components. Therefore, authors should highlight the essence of a sample clean-up for the following projects. In addition, they should emphasize the fact that one of the procedures to normalize such matrix effect in quantitation of the target analyte must be implemented (matrix-matched calibration, standard addition of the use of an appropriate internal standard).

: Thanks for the reviewer’s critical comments. We absolutely agree that the matrix can affect the signal since the recovery of the signal was 68-81% for ARV-110 and 75-78% for IS (Table 2). However, the matrix effect for ARV-110 and IS were consistent and reproducible for all the investigated concentrations, which is acceptable by the US FDA guidance [2]. Since the criteria by US FDA guidance recommends the CV of matrix effect should be lower than 15%, we tried to calculate the CV of matrix effect values, and calculated CV was found to be less than 15% for all QC levels of ARV-110 and IS. Accordingly, we have added the standard deviation and CV in the revised Table 2 and mentioned the CV in line 143-145. The SD values for decimal points were also corrected to be consistent with mean values in Table 3 and 4.

Moreover, after normalization with IS, the recovery of the signal in the matrix effect study was 87.06%–104.06% in rat plasma and 92.29%–104.83% in mouse plasma. These values indicate that the matrix does not affect the quantitation of ARV-110. Furthermore, in method validation, we established good calibration curve of ARV-110 in plasma (i.e., R=0.9921 for rat and R=0.9912 for mouse). Both standard and quality control (QC) samples were in the same matrix and internal standard was also used, resulting in the acceptable accuracy and precision in the validation. With simple protein precipitation, the lower limit of quantitation (LLOQ) was 2 ng/mL, which is sufficient to determine pharmacokinetic samples upto 48 hr.

In addition, in order to determine the cause of matrix effect, we performed an additional study about the matrix effect using PBS Buffer (pH 7.4) instead of plasma as a matrix. Notably, it was found that the decrease of intensity between two groups (i.e., PBS and rat plasma groups) was very similar (72.10 for PBS vs. 73.23% for rat plasma), suggesting that the matrix effect is likely due to increase of pH and/or salt, not the endogenous material (interferences) which may not be removed during the pretreatment process.

Finally, we agree with the reviewer that contamination of the instrument is possible and thus we have added the requirement of a sample clean-up (please see lines 149-154) as below.

“Since the instrument was possibly contaminated by the matrix components of plasma, a sample clean-up was performed prior to and after each sample injection using methanol 50% as washing solvent. In addition, the curtain plate surface of the mass spectrometer was washed with methanol 50% to remove any retained contaminants prior to starting following projects.”

  1. Please explain what the basis is of selecting pomalidomide as an internal standard because its molecular structure and molecular mass are different from than analyte. Any information on chemical properties?

: Thanks for the reviewer’s insightful comments. Pomalidomide and ARV-110 have a thalidomide group in their molecular structure (Figure 1). As a result, pomalidomide showed similar values to ARV-110 in matrix effect, recovery, and process efficiency tests. In addition, pomalidomide has the advantage of being a commercially available substance. For the above reasons, pomalidomide was used as an internal standard despite the difference in molecular weight.

  1. Chemical properties of analytes and the internal standard must be provided (logP, Pka).

: Thanks for the reviewer’s insightful comments. Based on the reviewer’s suggestions, we have added logP and pka value from the literature of pomalidomide (IS, line 230). However, there have not been any data of logP and pKa reported for ARV-110 in the literatures. Therefore, according to the reviewer’s suggestion, we have mentioned the molecular formula, molecular weight, and the predicted value of logP and pKa obtained from MarvinSketch version 22.7 software (please see line 228-229).  

  1. Line 125: please add the acceptable limits by citing the appropriate guidelines in “These values were within acceptable limits, in”

: We added the values (20% for LLOQ and 15% for other concentrations) with appropriate guidelines citation as suggested by the reviewer (please see line 130-131) as below.

“These values were within acceptable limits (i.e., 20% for LLOQ and 15% for other concen-trations), indicating that the ARV-110 analysis method was reproducible and reliable in rat plasma [12,13]”

References for rebuttal

  1. Agency, E.M. Guideline on bioanalytical method validation. EMEA/CHMP/EWP/192217/2009 2011.
  2. US-FDA. FDA Guidance for Industry: Bioanalytical Method Validation. US Department of Health and Human Services, FDA. Center for Drug Evaluation and Research, Rockville, MD, USA, in: https://www. fda. gov/downloads/drugs/guidances/ucm070107. Pdf 2018.
  3. Vuignier, K.; Schappler, J.; Veuthey, J.-L.; Carrupt, P.-A.; Martel, S. Drug–protein binding: a critical review of analytical tools. Analytical and Bioanalytical Chemistry 2010, 398, 53-66, doi:10.1007/s00216-010-3737-1.
  4. Kerns, E.H.; Di, L. Chapter 14 - Plasma Protein Binding. In Drug-like Properties: Concepts, Structure Design and Methods, Kerns, E.H., Di, L., Eds. Academic Press: San Diego, 2008; https://doi.org/10.1016/B978-012369520-8.50015-2pp. 187-196.
  5. Kim, M.-J.; Kwon, S.-H.; Jang, C.-G.; Maeng, H.-J. Determination of isoorientin levels in rat plasma after oral administration of Vaccinum bracteatum Thunb. methanol extract by high-performance liquid chromatography-tandem mass spectrometry. Biomedical Chromatography 2018, 32, e4188, doi:https://doi.org/10.1002/bmc.4188.
  6. Han, D.-G.; Cha, E.; Joo, J.; Hwang, J.S.; Kim, S.; Park, T.; Jeong, Y.-S.; Maeng, H.-J.; Kim, S.-B.; Yoon, I.-S. Investigation of the Factors Responsible for the Poor Oral Bioavailability of Acacetin in Rats: Physicochemical and Biopharmaceutical Aspects. Pharmaceutics 2021, 13, 175.
  7. Jin, H.-E.; Kim, I.-B.; Kim, Y.C.; Cho, K.H.; Maeng, H.-J. Determination of cefadroxil in rat plasma and urine using LC–MS/MS and its application to pharmacokinetic and urinary excretion studies. Journal of Chromatography B 2014, 947-948, 103-110, doi:https://doi.org/10.1016/j.jchromb.2013.12.027.
  8. Yoon, J.-H.; Nguyen, T.-T.-L.; Duong, V.-A.; Chun, K.-H.; Maeng, H.-J. Determination of KD025 (SLx-2119), a Selective ROCK2 Inhibitor, in Rat Plasma by High-Performance Liquid Chromatography-Tandem Mass Spectrometry and Its Pharmacokinetic Application. Molecules 2020, 25, 1369.
  9. Nguyen, T.-T.-L.; Duong, V.-A.; Vo, D.-K.; Jo, J.; Maeng, H.-J. Development and Validation of a Bioanalytical LC-MS/MS Method for Simultaneous Determination of Sirolimus in Porcine Whole Blood and Lung Tissue and Pharmacokinetic Application with Coronary Stents. Molecules 2021, 26, 425.
  10. Kim, S.B.; Lee, T.; Lee, H.S.; Song, C.K.; Cho, H.J.; Kim, D.D.; Maeng, H.J.; Yoon, I.S. Development and validation of a highly sensitive LC-MS/MS method for the determination of acacetin in human plasma and its application to a protein binding study. Arch Pharm Res 2016, 39, 213-220, doi:10.1007/s12272-015-0697-1.
  11. Waters, N.J.; Jones, R.; Williams, G.; Sohal, B. Validation of a Rapid Equilibrium Dialysis Approach for the Measurement of Plasma Protein Binding. Journal of Pharmaceutical Sciences 2008, 97, 4586-4595, doi:https://doi.org/10.1002/jps.21317.

Round 2

Reviewer 2 Report

The authors of this manuscript performed appropriately the revision process. I would suggest accepting this article for publication in Molecules.